# Aspects of Illness and Death among Roma—Have They Changed after More than Two Hundred Years?

**DOI:** 10.3390/ijerph16234796

**Published:** 2019-11-29

**Authors:** Michal Kozubik, Jitse P. van Dijk, Daniela Filakovska Bobakova

**Affiliations:** 1Department of Social Work and Social Sciences, Faculty of Social Sciences and Health Care, Constantine the Philosopher University in Nitra, 949 74 Nitra, Slovak; mkozubik@ukf.sk; 2Department of Community & Occupational Medicine, University Medical Centre Groningen, University Groningen, Deusinglaan 1, 9713 AV Groningen, The Netherlands; 3Olomouc University Social Health Institute, Palacky University in Olomouc, 771 11 Olomouc, Czech Republic; daniela.filakovska@upjs.sk; 4Graduate School Kosice Institute for Society and Health, Faculty of Medicine, Safarik University in Kosice, 040 01 Kosice, Slovak; 5Department of Health Psychology, Faculty of Medicine, University of Pavol Jozef Safarik in Kosice, 040 01 Kosice, Slovak

**Keywords:** illnesses, death, Roma, Slovakia, 1775, 2012

## Abstract

Augustini studied Roma and published reports in 1775–1776 on their illnesses and death. Our intention was to compare the features of these two topics described by him in the late 18th century with those in the present time. We studied Augustini’s work on illnesses and death in the past. The present qualitative study was conducted in 2012–2013 in the same geographical area in which Augustini lived and worked more than two hundred years ago, i.e., the Tatra Region in Slovakia; our findings were evaluated in 2017–2018. We carried out semi-structured interviews with more than 70 informants and organised two sessions of focus groups. Data were analysed using content analysis (Augustini) and an open coding process. Our findings suggest that illnesses in Roma are treated differently nowadays compared with 1775–1776. For example, the traditional forms of healing have completely disappeared in the area of investigation. We did not observe any differences in dying and death perceptions between the past and nowadays. Although data and knowledge on health disparities and related mechanisms exist, and much more about perceptions of Roma regarding illnesses is now known compared with 1775–1776, so far, this knowledge has not helped to design effective interventions to overcome them. Substandard living conditions in marginalised Roma communities have not significantly improved since 1775–1776, which may contribute to their higher morbidity and mortality also nowadays. Political and social consensus should lead to a comprehensive vision for enhancing the social situation and living conditions in segregated settlements, especially providing housing for the poorest classes and overcoming health disparities.

## 1. Introduction

Distinct health beliefs and practices among Roma related to illnesses and death have attracted attention from various fields of research [1,2,3,4,5,6,7,8,9,10,11,12]. The current research interest mostly focuses on topics such as the description of Roma health [13], Roma health behaviour [14,15], understanding Roma health [16] and its determinants [17] from the management of most common illnesses among the Roma population [18] and health-mediation programs [19] to comparisons with the non-Roma population regarding specific chronic illnesses [20,21,22,23,24,25,26]. Seemingly, some beliefs and practices are more stable than others, and some are more prone to change over time. How the opinions on illness and death of Roma from their perspective have changed after centuries has not been described so far, although a comparison on food and accommodation has been published [27]. We focused, therefore, on comparing two related issues, i.e., illnesses and death from the Roma perspective as they were described in the late 18th century and 250 years later.

In the past, very difficult living conditions connected with the nomadic way of life, low standards of personal hygiene and irregular eating habits were unfavourable factors determining health in Roma people [28]. In general, people maintained their health using a complex of medicinal practices which were based both on experience and on magic. Various herbs, garlic, onion, animal fat (from ducks, geese, dogs) and application of various objects on body parts (warmed stones, wet rags, cooked onion, herbal or fat compresses) were used to treat different kinds of respiratory, gastrointestinal and other illnesses. Folk healing is a combination of rational and irrational actions, which results from long-term observations of and learning about nature. In most cases, Roma knowledge in the area of phytotherapy corresponded with the knowledge used in the Slovak folk environment, and thus also in Roma living elsewhere [28,29].

Roma in the eastern part of Slovakia still believe that one of their dead relatives comes to take a dying person to the next world. They call the person *mulo poslancos* (dead envoy) or *guľi daj* (sweet mother); however, that person can be seen only by the dying, who can also see other dead relatives who gather by their bed or in the corners of the room. When the relatives see that there is no help for the dying person, they try to relieve his/her suffering. An example is that there must not be a pillow with feathers under the dying person’s head because then he/she cannot die. They let the person pass away in a bed and do not put him/her on the floor. Sometimes they undo women’s hair and take away their earrings, rings and ribbons. The reason is their belief in the easier departure of the spirit from the body [30].

To our knowledge, no comparison of attitudes to illnesses and dying in Roma between the 18th century and the present currently exists. In this study, therefore, we compared illnesses and death as depicted by Augustini ab Hortis in 1775–1776 with the contemporary state of affairs among Roma in the Tatra Region of eastern Slovakia.

## 2. Methods

First, we collated the information reported by Augustini ab Hortis on illnesses and death in the journal *Kaiserlich Königliche Allergnädigste Privilegierte Anzeigen aus Sämstlichen Kaiserl. Königl. Erbländer,* issued in the Austro-Hungarian Empire in 1775–1776 [31,32]. In order to maintain the highest possible compatibility of the research design with that of Augustini, we used research instruments based on Augustini’s work. Our study was designed on the basis of a qualitative method of data collection, the semi-structured interview. The analysis of the collected data was derived from the paradigm of the so-called new ethnography [33,34,35]. Our field study was conducted in the same region in which Augustini lived and worked in the 18th century.

### 2.1. Samples

Augustini lived in the Tatra Region in the north-eastern part of Slovakia [36,37], where a subethnic group called Rumungre Roma live, the same group which we studied. The data were collected in five localities of the Poprad district: Hranovnica, Spisske Bystre, Vikartovce (segregated settlements), Kravany (a separated concentration) and the town of Poprad itself. The Poprad district has a relatively high share of Roma population, corresponding to more than 30% [38]. We co-operated with four key informants, interviewed more than 70 Roma people, and organised two focus groups. Each group consisted of 20 informants. The participants in both groups were chosen randomly and met certain criteria: they had to identify themselves as Roma and had to live in a segregated settlement. All of them lived in extreme poverty and were dependent on welfare benefits. We recorded more than 1700 min of interviews and took many photographs. All the addressed people confirmed their ethnicity and voluntarily consented to participation in the study in audio recordings.

### 2.2. Data Collection

Prior to the fieldwork, we prepared a detailed history, which preceded the semi-structured interviews. It included the content analysis of Augustini’s work regarding illnesses and dying. This was followed by a visit to each settlement. An intensive ethnographic study with a stay in the settlements was conducted in the summer months of 2012–2013. We recorded the data in two ways: by taking research notes in a fieldwork logbook and by making audio recordings. Before our field research, we arranged a detailed anamnesis based on a semi-structured interview with each informant. Within conversations regarding illness and death, we focused on two leading questions comprising the main aim of the study: “How do Roma perceive and take care of their health and how did they perceive it in the past?” and “What do the people from the community think about death and dying compared to the past?” The essential attribute in the study was a person or a key informant who enabled us to share their and their family’s everyday reality in separated settlements. Every day, we went into the field from their dwellings. We started in the poorest parts of the localities; within those, we chose the first informants randomly. After the first visit in these parts of the area, it was much easier to find other informants; it is not usual to see any non-Roma in the poorest zones. Many Roma people wanted to meet us and shared their daily routine with us. The population of the settlements is heterogeneous with regard to their socioeconomic status. It consists of people living below the poverty line, a middle class and rich family clans. Many of the first informants gradually introduced us to their extended families, and our sample increased in number through the “snowball effect”. The mapping also included the evaluation of the collected data; the study extended between January 2017 and June 2018.

### 2.3. Analyses and Reporting

Besides his own observations, Augustini acquired information in 1775 about illnesses and death in Roma from the contemporary literature and older manuscripts. He quotes dozens of authors, citing them accurately and as developed references, which was unusual at that time. Augustini approaches the opinions of the authors as a critical research scientist, refuses romantic and inexpert opinions and focuses on those he considered to be correct, producing a kind of narrative review. He supplements the data from the literature and sources with descriptions of the then life of Roma from his observations and using information from his contemporaries [31].

Our analytic approach was inspired by the opinions and methodology of social sciences according to Weber, who emphasises the avoidance of the researcher’s excessive subjectivity and judgements [39,40,41]. Geertz, a founder of the school of so-called new ethnography, was significantly influenced by his teachings [33,34,35]. The evaluation of the data was conducted through simple, open coding, followed by the development of central categories. The same categories for both historical periods were compared. For the purpose of this study, we focused on all parts of the interviews where the topics of illness and death arose. We report the most important statements in verbatim transcriptions of the recordings; they were then translated into English by the authors of the study.

### 2.4. Research Ethics

In our research, neither animals nor plants were studied. Human beings from 1775 to 1776 were not studied by us, and human beings in 2012 and 2018 were studied in line with the Helsinki Declaration. All of the participants agreed to their participation in the study. Their informed consents were obtained and archived through audio recordings.

## 3. Results

We focused on two areas: illnesses and death. Augustini, however, conceptualised these two variables in one. To maintain compatibility, we decided to proceed in the same way. For better understanding, we provide the verbatim statements of the study participants. The results are stated individually for each historical period for better clarity.

### 3.1. Roma Illnesses and Death in the Late 18th Century

Augustini’s original version of the treatise on illnesses and death is rather brief (Figure 1). In an introductory sentence, he emphasises the healthy and strong physical features of Roma. He is surprised that despite the risky life they live, morbidity is lower than in the majority population. On the other hand, Augustini also states that illnesses are present in Roma too. Among typical childhood illnesses, he mentions smallpox and measles. He has no knowledge about the mortality of Roma children related to the stated infectious illnesses as being high or not. Roma suffer from cataracts and fever only seldom, and gout seems to be very rare. He observes that a plague epidemic “sneaks into their residential houses rather than into their tents”. He notices problems with sight and attributes them to the daily presence of fire and smoke in their tent dwellings. During their lives, however, they suffer from illnesses only seldom. However, if Roma suffer any illness, they avoid doctors. They do not ask them for advice nor do they use any drugs from a pharmacy. They leave the cure to lucky coincidences and herbal remedies. For example, they use saffron and saffron soup to deal with stomach and digestion problems. Because they believe that bloodletting helps horses, they use it too. Unfortunately, Augustini does not mention what kind of illnesses were treated this way. Perhaps, this belief was strengthened because the idea of restoring the body’s balance by bloodletting in the sense of Galen’s four humours theory was quite normal in those days [42].

Augustini does enrich our knowledge with special information about the dying process among Roma. As in the majority population, Roma also tend to fall ill before death, and their life ends after a period of lying in bed, if no accident has happened. They are afraid of death, and their preparation for it relates to their particular religious beliefs. The death of a Roma community member is followed by crying and lamenting, and women and men tear out their hair, particularly if a close person dies, such as a wife, husband or children. Those who die by natural causes are buried in the same way as in the majority population. There are differences only in the lamenting, screaming and grief manifestations. When an official *vajda* (head of the community) dies, great honours are paid to his body. According to Augustini, it is not unusual for Roma community members to die a violent death on the gallows or the breaking wheel. In such cases, there is even more crying and grief at the funerals, which Augustini perceives as theatrical and comedy-like manifestations. However, he sensitively states that a funeral is a very sad event for a family. Augustini states a specific case and cites the local press (unfortunately, he does not state the source) when Roma addressed the officials who were present during the execution. Their plea for their relative who was sentenced to death by hanging was: *“Gentlemen, do not violently force a man towards something for which, as you can see, he does not have the tiniest will and liking!”* Augustini depicts creativity and ideas as frequent in attempts to avoid the performance of capital punishment. In the conclusion, he states that no suicide attempts were found among Roma: *“No Roma person would ever finish his life on his own and intentionally only because of worries. His life is too kind and pleasant for him. Despite many worries, they are never overcome with despondency and melancholy, and even in the greatest need they are happy, in good mood and they die only when they have to die.”*

### 3.2. Roma Illnesses and Death in the 21st Century

Illnesses are perceived as common in the community, and cancer in particular is perceived as a common cause of death. Roma visit the doctor if necessary. The problem are the financial costs: travel expenses and access to health care. The poorest families have these problems very often. More attention is paid to the children, and women are mostly the ones who take care of health issues in the families. For men, they visit the doctor only for something really serious. Life expectancy of Roma compared to non-Roma is low, and Roma older than 50 are considered to be old. We found no forms of healing based on the use of herbal therapy, nor did we observe the use of bloodletting. Here, we witnessed a great difference with respect to the use of phytotherapy stated in the Introduction [28,29,43]. The only alternative way of treatment we identified was using dog fat to treat bronchial illnesses (instead of the formerly used fat from the European badger (*Meles Meles*), as poaching is a crime in Slovakia). We found this only after a longer period spent in the settlement. It is taboo because, in accordance with the legislation, animals should not be tortured in any way.

At the time of our stay in the settlement, several families were affected by tuberculosis: “Now, tuberculosis is here. Most often, people die of cancer, both young and old ones. Also young people die here, otherwise they live for up to 70 years. Here, when you are 50, they say that I am already old.” (Roma woman, 49). At the time of our fieldwork, there was a health education assistant coming from the Roma community working in the settlement within the Healthy Communities project [44]: “Tuberculosis—its treatment regimen is problematic; bad conditions and so on are thought to be its determinants; cancer; colon cancer… Our job was that we walked around and warned people about the spreading of diseases and about hygiene, what might happen without it. We spread that education among the Roma community; we should have more of that. And we urged them to have their vaccinations on time.” (Roma man, 32). The members of the Maranata Christian Mission (more than 1000 Roma believers and rising in Slovakia [45], one of two central churches in Poprad), which is based on the example of pastoral activities in the Afro-American community, have their own opinion on illnesses. “Illnesses … some people cause their own illnesses … they smoke, drink and so on. But there are also hereditary illnesses. Also there are illnesses that cannot be cured, but I think that faith is also necessary there. When I am sick, I do not want drugs. I have not taken any drugs for five years. I pray. Lord, you live in me, give me strength. I have my God, I have my faith, I know Him. I do not want to be addicted to drugs, but people are addicted.” (Roma pastor, 35). They believe in an afterlife. In the community, a specific role is also played by Pentecostalism, which also perceives illnesses through religious optics.

Fear of death and a strong connection to religiosity and animism are still observable nowadays: “Every single person is afraid of death, I believe in an afterlife. If it was not so, why do I believe in God; we would not believe in God.” (Roma woman, 49). The customs related to dying, death and burying were also described: “Funerals are classical here as they are in the majority. In the past, it used to be different; a person who died was in the house, yes, there were no morgues, so it existed then, mulos and haunting. Nowadays, it is the opposite, death is normal. Roma believe that mulos exist.” (Roma man, 52). Believing in ghosts, so-called mulos, is still present. Roma, however, do not die only at home but, more often, in hospitals, in an institutional environment. Moreover, death has become taboo as in the majority population. Crying and lamenting, however, in contrast to the majority population, are still very temperamental (Figure 2).

## 4. Discussion

Our intention was to compare illnesses and the process of dying among Roma as described by Augustini in 1775–1776 and as they occur presently. Current studies are mostly oriented towards the characterization of Roma health, behavior and determinants and their comparison with non-Roma [13,14,15,16,17,18,19,20,21,22,23,24,25,26]. The comparison of the states of these topics over time is missing. Thus, our aim was to compare the state of two topics, illness and death, in two different periods of time, i.e., the 18th and the 21st centuries from the Roma point of view. In comparison with the past, folk healing seems to be disappearing. The attitude to dying does not differ from that of the majority population, although elements of animism and belief in *mulos* are still present. The members of the Christian Maranata Church are an exception: they believe in an afterlife and refuse any elements of demonising.

### 4.1. Illnesses

Illnesses are perceived by Roma as common in the community. Roma are affected mostly by infectious illnesses such as tuberculosis and hepatitis, which are to some extent associated with their low hygienic standard of living [46,47]. Cancer from the Roma perspective is perceived as a cause of death among young and old equally. In reality, about 1% of all who die from cancer are aged 19 years or younger [48,49]. Augustini was surprised that, despite the risky life they lived, morbidity was lower than in the majority population. At the present time too, some lay persons believe that Roma health is more resistant compared to that of non-Roma people. However, the European Commission estimates that in Slovakia, life expectancy (LE) is 55.3 years for Roma men and 59.5 years for Roma women [50], which is approximately 10 years less than the LE of the general population. According to Infostat, LE is 64.4 years for men and 71.6 years for women in Slovakia [27,51]. At present, as in the past, there is reluctance among Roma to visit doctors, and they tend to avoid the use of medicines. The reason lies not only in financial barriers or problems with access to health care but also in mistrust of state authorities (e.g., school, civil service) [52,53].

We did not find practice of any traditional medical procedures described by Augustini in the 18th century in our research area. Previous studies focusing particularly on descriptions of folk healing are rare and describe Roma living in a different geographic area of Slovakia [28,29,30,43]. For decades, these works have encouraged mythical and romantic views of herbalism and natural healing sources. The current situation in the visited localities is, however, completely different. We did not meet a folk healer among them. Augustini observed that they also used bloodletting. They believed that, since this method helped horses, it could also help them. It seems that such practices have completely disappeared, or at least we did not find any evidence of them.

### 4.2. Death and Dying

The approach to dying and death does not show any significant differences in comparison with that of the majority population, except for the crying and lamenting among Roma, which was similarly described by Augustini. Temperamental manifestations of grieving during burials are still present. There is still a difference from the majority at present in the strong belief among Roma in an afterlife and in ghosts of the dead, as Augustini described in 1775. A special denomination, the Maranata affiliates, refuse everything that deviates from the Holy Scripture and their belief in Jesus Christ. They repudiate the worshipping of idols, decorations of dwellings with pictures of the saints and performing magic. Augustini describes Roma dying very briefly: they died due to illness or to old age on a simple bed placed on the ground. Unlike in the past as described by Augustini, at present Roma mostly die in a hospital environment, as a result of a natural process of medicalisation. If Roma die in the settlements, unpleasant situations faced by emergency medical service workers may occur: *“When I arrived in the settlement, I saw there was no help for that man. But everybody there screamed, so I started to resuscitate him, and we brought him into the ambulance. We were scared they would hurt us if we had not resuscitated him.”* (paramedic, non-Roma man, 24).

### 4.3. Strengths and Limitations

Through our content analysis of Augustini’s work, we tried to provide a complete view of illnesses and death in Roma in Augustini’s period. Furthermore, we compared the past with the present situation in the same geographic area where Augustini lived more than 200 years ago. The Roma population varies in this region regarding the places where they have settled. For this reason, we tried to cover all types of Roma communities in the High Tatra region: people living in separated and poor localities as well as Roma who live in the town of Poprad.

We used individual interviews to explore personal experiences and focus groups to examine opinions and beliefs about the phenomena of death and illness among Roma. Although focus groups and individual interviews are different data collection methods, their combination can be advantageous to researchers, as complementary views of the phenomenon in question [54,55]. Our findings shed light on how the behaviours among Roma related to illness and death have changed over time. Some limitations should be mentioned as well. As there are substantial differences in the sociocultural norms between two subethnic groups of Roma in Slovakia, *Vlachiko Roma* and *Rumungre Roma*, our findings cannot be generalised without any restrictions to the whole Roma ethno-national minority in Slovakia but only apply to a narrowly defined geographic area, specifically, the High Tatra region.

### 4.4. Recommendations

In the poorest localities, tuberculosis, which was successfully suppressed in the past, has started occurring again. Furthermore, any enhancement of the standard of living depends on the particular settlement, the areas where houses are built and the possibility of housing improvement. In order to improve the health outcomes for the population living in separated and segregated Roma settlements, interventions should be focused on social determinants of health, including housing, infrastructure and access to piped water, health care, education and employment. The existing Strategy of the Slovak Republic for Roma integration [56] has met these requirements only partially. There are still a lot of localities where the standard of living needs to be improved. The political effort should aim at covering the basic needs of poor Roma, especially regarding health and housing. The latter is an indispensable condition for ensuring good health conditions for the poorest Roma.

Roma are still influenced by religiosity [45,57,58]. They welcome any interest shown by churches or religious associations. No change in the approach to life in the context of religiosity, however, has produced any real social change [45]. Exploring the reasons for this could be a subject for further research. Social change is one of the main goals of the social work profession [59]. Evidence is lacking regarding the empowerment of the poorest Roma in Slovakia, and the issue of how to empower them for social change could be a crucial research question for social work in Slovakia.

## 5. Conclusions

We compared attitudes to illness, death and dying in the past with those in the present in the Roma settlements and communities in the Tatra Region. The traditional form of healing has practically disappeared. In the late 18th century, the living conditions were not much different as far as hygienic standards are concerned, which means that the geographically close non-Roma population also lived in much worse conditions compared with how people live nowadays. Thus, the causes of illnesses in Roma then and now are not much different. On the other hand, the living conditions of the majority population have improved, and this is widening the health gap. Nowadays, in the segregated settlements, there are illnesses which are associated with low standards of living and hygiene. The poorest Roma may have no access to piped water [38]. Medicine is still not successful in exterminating tuberculosis. Health literacy among the inhabitants of the settlements has been enhanced by health education assistants’ activities, but they can improve health-related outcomes only partially, as their activities are not accompanied by a significant improvement in living conditions [16].

Attitudes to dying and respect for the dead do not differ substantially from those of the majority population, except that Roma, as before, still strongly believe in ghosts, *mulos*. Attitudes to dying and death among Roma nowadays do not differ much from those of the past, as described by Augustini. The only difference we identified is that dying at home is less prevalent than it used to be, as dying in hospital is more prevalent nowadays. Our conclusion has to be that some aspects of life for Roma related to illness and death have changed, whereas some of them persist, and the main factor for change appears to be the interaction between Roma communities and the majority population [37].

## Figures and Tables

**Figure 1 ijerph-16-04796-f001:**
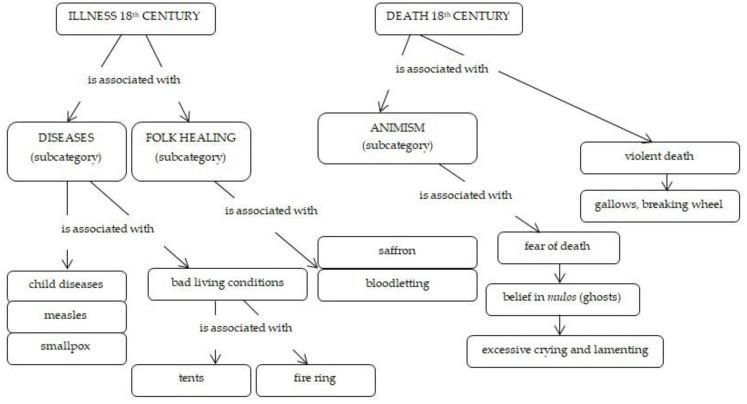
Illness and death of Roma in the 18th century.

**Figure 2 ijerph-16-04796-f002:**
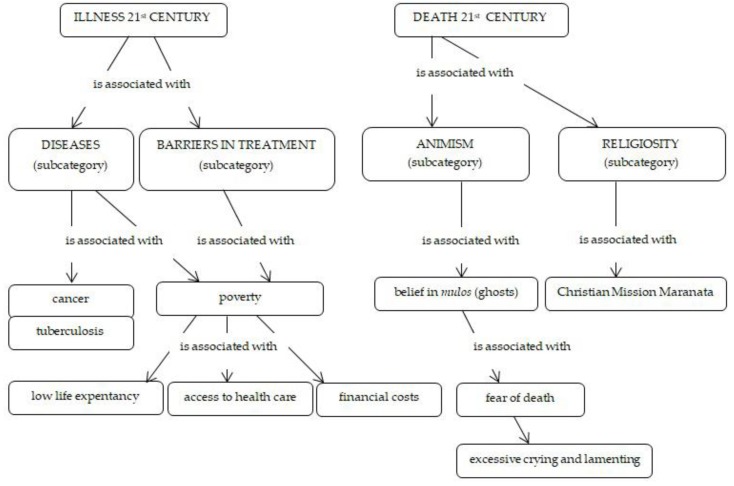
Illness and death of Roma in the 21st century.

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
