# Peer review of "Aspects of Illness and Death among Roma—Have They Changed after More than Two Hundred Years?"

_ijerph, 2019, doi:10.3390/ijerph16234796_

Round 1
Reviewer 1 Report
The present study was to compare the difference between different time of Augustini lived and worked related to illness and death during two hundred years, based on qualitative study of focus study. This is an unique historical medicine study which provide useful information in modern healthcare system. Many issues needed to be clarified in this study:
1.The author conducted this study to collect data from 70 interviewees from two focus groups. how to evaluate the consistence of these two focus groups and the sample selection process should describe in the content.
2. The results provide few detail information of interviewings instead of authors's subjective information. There were 70 interviewees should provide much information on the research themes.
3. There are still have many limitations on this study such as data collection, varied samples...needed to put in this section.
Author Response
please have a look in the attached file

Reviewer 2 Report
The article is well built and it is formed by the main sections that are expected in a scientific article. The Introduction mentions main topics analysed in previous studies in this same field of knowledge but there is not an extended discussion on them. This is an aspect that could be improved in the whole article, to provide more detail of the article’s background.
The design of the article is appropriate, with a qualitative method within a new ethnography paradigm. This article is based on a comparison with the data obtained from a previous study of Augustini in the 1775-1776 period and data from a current period obtained from the fieldwork implemented by the authors. The terms of this comparison need to be further explained in order to understand how this comparison has been done. For example, in line 126 of the text it is mentioned the use of categories to analyse the qualitative data but is not explained if the authors used the same categories to analyse the two periods or not. Further explanation on that would be useful.
It is relevant to mention that the results and the discussion sections are very well structured. These sections provide evidences on the need to deeper in the structural barriers which affect the Roma in terms of health and wellbeing. It is also appreciated the identification of the limitations of this study, and of potential topics to be further explored. In addition it is also relevant the social impact approach of the paper, highlighting recommendations for the improving of the living conditions of the Roma communities. The conclusions are also providing interesting elements for the overcoming of the conditions of poverty which still affect many Roma people.
Author Response
Please have a look in the attached file
